# Different adjuvanted pediatric HIV envelope vaccines induced distinct plasma antibody responses despite similar B cell receptor repertoires in infant rhesus macaques

Stella J. Berendam[1‡], Papa K. Morgan-Asiedu[1‡], Riley J. Mangan[1], Shuk Hang Li[1], Holly Heimsath[1], Kan Luo[1], Alan D. Curtis, II[2], Joshua A. Eudailey[1,3], Christopher B. Fox[4,5], Mark A. Tomai[6], Bonnie Phillips[2], Hannah L. Itell[1], Erika Kunz[1], Michael Hudgens[7], Kenneth Cronin[1], Kevin Wiehe[1], S. Munir Alam[1], Koen K. A. Van Rompay[8], Kristina De Paris[2], Sallie R. Permar[1,3], M. Anthony Moody[1‡], Genevieve G. Fouda[1‡]*

1 Duke Human Vaccine Institute, Duke University Medical Center, Durham, North Carolina, United States of America, 2 Department of Microbiology and Immunology, Children's Research Institute and Center for AIDS Research, School of Medicine, University of North Carolina at Chapel Hill, Chapel Hill, North Carolina, United States of America, 3 Department of Pediatrics, Weill Cornell College of Medicine, New York City, New York, United States of America, 4 Infectious Disease Research Institute (IDRI), Seattle, Washington State, United States of America, 5 Department of Global Health, University of Washington, Seattle, Washington State, United States of America, 6 3M Center, 3 M Drug Delivery Systems, St. Paul, Minnesota, United States of America, 7 Department of Biostatistics, University of North Carolina at Chapel Hill, Chapel Hill, North Carolina, United States of America, 8 California National Primate Research Center, University of California at Davis, Davis, California, United States of America

‡ SJB and PKM shared first authors. MAM and GGF shared last authors.
* genevieve.fouda@duke.edu

## Abstract

Different HIV vaccine regimens elicit distinct plasma antibody responses in both human and nonhuman primate models. Previous studies in human and non-human primate infants showed that adjuvants influenced the quality of plasma antibody responses induced by pediatric HIV envelope vaccine regimens. We recently reported that use of the 3M052-SE adjuvant and longer intervals between vaccinations are associated with higher magnitude of antibody responses in infant rhesus macaques. However, the impact of different adjuvants in HIV vaccine regimens on the developing infant B cell receptor (BCR) repertoire has not been studied. This study evaluated whether pediatric HIV envelope vaccine regimens with different adjuvants induced distinct antigen-specific memory B cell repertoires and whether specific immunoglobulin (Ig) immunogenetic characteristics are associated with higher magnitude of plasma antibody responses in vaccinated infant rhesus macaques. We utilized archived preclinical pediatric HIV vaccine studies PBMCs and tissue samples from 19 infant rhesus macaques immunized either with (i) HIV Env protein with a squalene adjuvant, (ii) MVA-HIV and Env protein co-administered using a 3-week interval, (iii) MVA-HIV prime/ protein boost with an extended 6-week interval between immunizations, or (iv) with HIV Env administered with 3M-052-SE adjuvant. Frequencies of vaccine-elicited HIV Env-specific memory B cells from PBMCs and tissues were similar across vaccination groups (frequency

**Data Availability Statement:** All relevant data are within the paper and its Supporting information files.

**Funding:** This work was supported by grant P01 AI117915 from the National Institute of Allergy and Infectious Diseases (NIAID) to KDP and SRP, and P51OD011107 (Office of Research Infrastructure Program, Office of The Director, NIH) to California National Primate Research Center (CNPRC). SJB is supported by the Interdisciplinary Research and Training Program in AIDS (5T32AI007392-32) to Duke University from NIAID.

**Competing interests:** The authors have declared that no competing interests exist.

range of 0.06–1.72%). There was no association between vaccine-elicited antigen-specific memory B cell frequencies and plasma antibody titer or avidity. Moreover, the epitope specificity and Ig immunogenetic features of vaccine-elicited monoclonal antibodies did not differ between the different vaccine regimens. These data suggest that pediatric HIV envelope vaccine candidates with different adjuvants that previously induced higher magnitude and quality of plasma antibody responses in infant rhesus macaques were not driven by distinct antigen-specific memory BCR repertoires.

## Introduction

In 2019, 85% of the estimated 1.3 million pregnant women living with HIV-1 globally received antiretroviral drugs to prevent transmission to their children [1]. While the implementation of antiretroviral prophylaxis has significantly decreased the global frequency of mother-to-child transmission (MTCT) of HIV-1, issues of maternal adherence to antiretroviral therapy (ART) [2, 3], development of ART-resistant viruses [4], and insufficient coverage of ART in some of the hardest-hit areas globally have limited the effectiveness of ART [5]. Furthermore, women with acute HIV-1 infection in late pregnancy or during the breastfeeding period are less likely to be diagnosed and receive treatment to prevent MTCT [6]. Thus, despite advancements in therapy, breast milk transmission still accounts for approximately 50% of pediatric HIV infections [7, 8]. Additional prevention strategies, such as a pediatric HIV-1 vaccine, are therefore critically needed to eradicate breast milk transmission of HIV-1.

Early efforts in HIV-1 vaccine development focused on the humoral arm of the immune system as other vaccines that successfully prevent viral diseases relied on antibodies for protection [9]. However, early-phase HIV-1 vaccine studies using recombinant HIV-1 envelope (Env) proteins showed no efficacy [10–12] with the exception of the RV144 trial that demonstrated moderate vaccine efficacy of 61% during the first year and an overall efficacy of 31% at 3.5 years after vaccination [13, 14]. Interestingly, non-neutralizing IgG that targeted the HIV-1 Env variable loops 1 and 2 (V1V2) were identified as correlates of protection from the RV144 study; meanwhile HIV-1 Env-specific IgA plasma antibodies were associated with lack of protection [15]. This finding reinvigorated the optimism that an effective HIV-1 vaccine is attainable. A phase 2b/3 study (HVTN 702 or Uhambo) in South Africa, which used a pox vector prime-protein boost vaccine regimen similar to RV144, albeit with distinct vaccine strains and a different adjuvant, did not reproduce the results from the RV144 study and showed no efficacy [16]. Nevertheless, these studies demonstrated that HIV-1 vaccine immunogens could induce robust levels of V1V2 IgG antibodies as well as polyfunctional CD4$^+$ T cell responses as surrogate of possible protection [15, 17].

To date, only a few previous pediatric HIV-1 vaccine trials have been conducted and these trials demonstrated that immunization with recombinant subunit HIV gp120 vaccines (PACTG 230) or with canarypox vectors expressing HIV antigens (PACTG 326, HPTN 027) are safe and immunogenic [18–21]. Importantly, HIV-1 infant vaccination with MF-59 adjuvanted HIV gp120 was able to generate robust and durable Env-specific IgG responses including anti-V1V2 IgG responses with low levels of Env-specific IgA responses [22]. Similarly, neonatal rhesus macaques are capable of developing robust virus-specific humoral and cellular immune responses after SIV vaccination [23, 24]. These clinical and preclinical studies demonstrate the feasibility of initiating HIV-1 immunization within the first few days of life.

Our previous studies in non-human primate infants have indicated that several factors can modulate the quality of the vaccine-elicited antibody response in infants [25, 26]. Notably, we observed that extending the interval between immunizations and the use of toll-like receptor (TLR) agonist adjuvants can enhance the magnitude and breadth of the cellular and humoral responses [26, 27]. However, the influence of distinct HIV vaccine regimens and adjuvants on the developing infant B cell repertoire has not been studied and the relationship between the magnitude and quality of HIV-1 vaccine-elicited responses and immunogenetic characteristics of the infant B cell repertoire remains unclear. Taking advantage of archived plasma samples obtained from completed preclinical studies [25, 26], we assessed the antigen-specific B cell repertoire in HIV vaccinated infant rhesus macaques. Our results indicated that while the magnitude and quality of plasma antibody responses induced by pediatric HIV vaccine regimens with different adjuvants, the B cell repertoire profiles were not distinct between different vaccine regimens in infant rhesus macaques.

## Methods

### Animals

A total of 19 Simian Immunodeficiency Virus (SIV)-negative and type D retrovirus-negative newborn Indian-origin rhesus macaques (*Macaca mulatta*) were hand reared in the nursery of the California National Primate Research Center (CNPRC, Davis, CA) as previously described [25]. Animals were reared in accordance with the American Association for Accreditation of Laboratory Animal Care Standards, the guidelines of the Guide for the Care and Use of Laboratory Animals of the Institute for Laboratory Research, National Research Council, and the International Guiding Principles for Biomedical Research Involving Animals. All protocols were assessed and approved by the University of California at Davis Institutional Animal Care and Use Committee prior to beginning the study. Animals were randomly assigned to groups and anesthetized for vaccinations and sample collection as previously reported [25, 26].

### Immunization regimens and study designs

Immunization regimens and archived samples used in this study for all 4 animal groups are summarized in Fig 1 and detailed study designs were previously published [25, 26]. Briefly, infants in group 1 (HIV Env+adjuvant, n = 5) were vaccinated at 0, 3 and 6 weeks of age intramuscularly (IM) with 5 x $10^8$ infectious units [IU] of MVA/SIV gag/pol, with 15 μg of C.1086 gp120 administered IM in Span85-Tween 80-squalene (STS) adjuvant, and with 200 μg of C.1086 gp120 administered intranasally (IN) in Toll-like receptor 7 and 8 (TLR7/8) agonist, R848 adjuvant [25]. Infants in group 2 (Co-administration, n = 5) were vaccinated at 0, 3 and 6 weeks of age with similar regimen as group 1 with addition of the MVA-HIV Env (5 x $10^8$ IU, IM). Infants from these two groups were followed for 19 and 15 weeks, respectively, after which they were euthanized to analyze vaccine-induced tissue responses. Infants in group 3 (Extended Interval, n = 4) received the same vaccine regimen as the co-administration group but were immunized at 0, 6 and 12 weeks and then boosted at 32 before euthanasia at week 35 [25]. Infants in group 4 (3M-052-SE, n = 4) were immunized at 0, 2 and 6 weeks of age with a combination of HIV.C.1086C gp120 and TV1 gp120 (15μg each, IM) in TLR7/8-based adjuvant 3M-052 formulated in stable emulsion (3M-052-SE) and was challenged with SHIV at week 15 [26].

### Collection and processing of blood and tissue specimens

Whole blood, plasma, and peripheral blood mononuclear cells (PBMCs) were collected before each immunization and thereafter biweekly throughout the study as previously described [25,

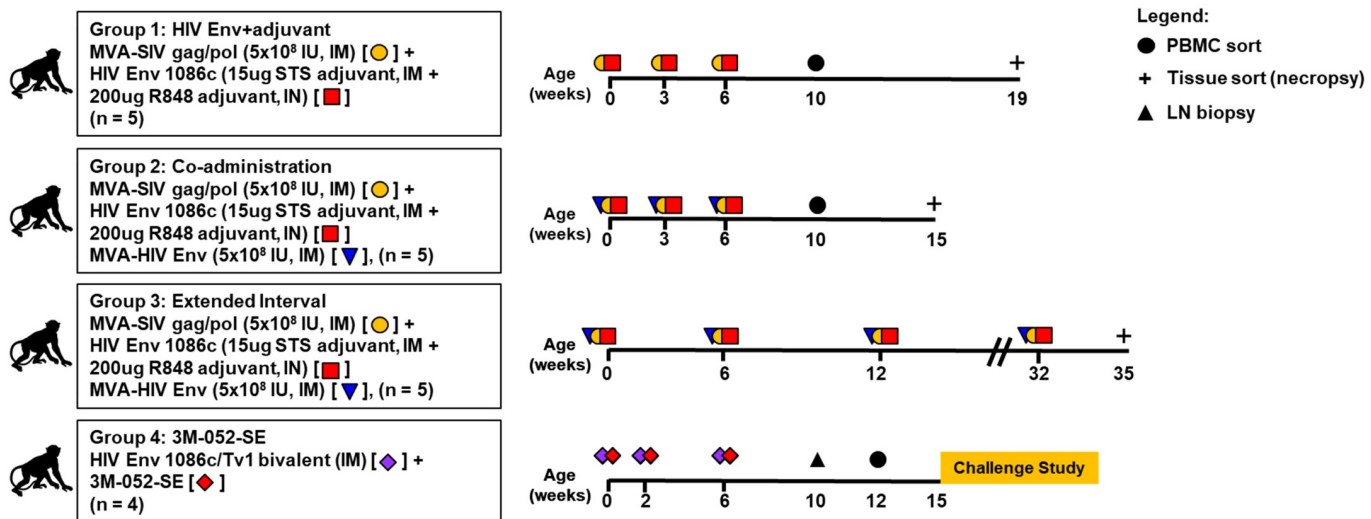

**Fig 1. Animal study design and immunization schedule.** Infant rhesus macaques from four immunization schedules were included. HIV Env+adjuvant group was immunized with MVA-SIV gag/pol and HIV Envelope (Env) 1086c adjuvanted with 15 μg STS, intramuscular + 200 μg R848, intranasal. Co-administration group was immunized with MVA-SIV gag/pol, HIV Env 1086c adjuvanted with 15 μg STS, intramuscular + 200 μg R848, intranasal), and with MVA-HIV Env. Extended interval group was immunized with similar immunogens as Co-administration group with longer immunization intervals. 3M-052-SE group was immunized with HIV Env 1086c/TV1 bivalent adjuvanted with 15 μg/15 μg+3M-052-SE, intramuscular. Plus symbol (+) denoted necropsy, IM denoted intramuscular, IN denoted intranasal, and IU denoted infectious unit.

26]. Archived peripheral blood and mononuclear cells (PBMCs) reported in this study were collected at week 10–12 for the HIV Env+adjuvant, the Co-administrationthe 3M-052-SE groups. There were no archived PBMC samples available at week 10 or 12 for the Extended Interval group and no PBMCs were sorted for this group. Archived tissue samples including spleen, lymph nodes (LNs; axillary, mesenteric, submandibular, cervical, submental, and retro-pharyngeal), and intestinal tissues (colon and ileum) were used for preparation of mononuclear cell (MNC) suspensions at necropsies for HIV Env+adjuvant, coadministration, and extended interval groups. Meanwhile archived LN biopsies from week 10 were used for preparation of MNC suspensions in the 3M-052-SE group. The 3M-052-SE group was part of a challenge study and therefore no necropsy tissues were available for antigen-specific B cell sorting. Summary of archived samples available for antigen-specific B cell sorting in this study are listed in Table 1.

## Single-cell flow cytometry sorting of antigen-specific memory B cells

Due to limited availability of archived samples, antigen-specific memory B cell sorting was performed in 15 of 19 infants (Table 1). Summary of percent antigen-specific memory B cells from archived samples of individual infants and sample types are listed in Table 1 and representative flow cytometry analyses for different sample types from each vaccine group are listed in S1A–S1J Fig. Briefly, PBMCs and tissue MNC suspensions were treated with 5 μM Chk2 inhibitor II or 2-[4-(4-chlorophenoxy)phenyl]-1H-benzimidazole-5-carboxamide (Sigma) prepared in final volume of 1% bovine serum albumin (Sigma-Aldrich) in 1X phosphate buffered saline (PBS, Sigma). PBMCs and MNC suspensions were blocked with 6.25 μg/ml anti-human CD4 antibody (BD Biosciences) at 4˚C for 15 min followed by staining with a panel of fluorochrome-conjugated antibodies to identify antigen-specific memory B cells as described by gating strategy (S1A–S1J Fig). Briefly, lymphocytes were gated on singlets and live cells based on Aqua vital dye (Invitrogen), exclusion of T cells (CD3-PerCP-Cy5.5, clone SP34-2,

**Table 1. Frequency of Env-specific memory B cells in sorted tissues and PBMCs of envelope-vaccinated infant monkeys.**

| Group | Animal ID | Percent (%) Ag+ Memory B Cells | | | | |
| --- | --- | --- | --- | --- | --- | --- |
| | | Spleen | Axillary LN | Oral LN¶ | Colon/Ileum | PBMC W10/12 |
| HIV Env+adjuvant | 45519 | ND | ND | ND | ND | 0.84 |
| | 45521 | 0.27* | ND | 0.17* | ND | ND |
| | 45522 | 0.1* | ND | 0.07* | ND | 1.72 |
| Co-administration | 45038 | ND | ND | ND | ND | 0.36 |
| | 45047 | ND | ND | ND | ND | 0.16 |
| | 45069 | ND | ND | ND | ND | 0.15 |
| | 45083 | 0.06 | ND | 0.32* | ND | 0.14 |
| | 45091 | 0.06 | 0.97 | ND | ND | 0.23 |
| Extended Interval | 45435 | 0.4* | ND | 0.49* | ND | ND |
| | 45441 | 0.08* | ND | 0.33* | 0.19* | ND |
| | 45448 | ND | ND | ND | 0.17* | ND |
| 3M-052-SE | 45838 | ND | ND | ND | ND | 0.11 |
| | 45840 | ND | ND | ND | ND | 0.11 |
| | 45847 | ND | ND | ND | ND | 0.06 |
| | 45851 | ND | ND | ND | ND | 0.06 |

*Average of 2–4 independent Ag+ B cell sorting experiment,

¶Oral LN consisted of axillary, mesenteric, submandibular, cervical, submental, and retropharyngeal LN, ND = No sort performed.

BD Biosciences) and monocytes/macrophages (CD14-BV570, clone M5E2, BioLegend; CD16-Phycoerythrin-Cy7, clone 3G8, BD Biosciences), followed by selection of memory B cells by positive expression of CD20 and CD27 (CD20-FITC, clone 2H7; CD27-APC-Cy7, clone O323, both BioLegend) but not immunoglobulin D (IgD-PE, Southern Biotech) with double specificity to both HIV C.1086 Env hooks (BV421-gp120 C.1086 and AF647-gp120 C.1086, both generated in-house).

## Polymerase chain reaction (PCR) amplification of immunoglobulin (Ig) $V_H$ and $V_L$ genes

The sorted single-cell antigen specific memory B cells $V_H$ and $V_L$ genes were amplified by nested PCR as previously described [28–30] followed by Sanger sequencing of the purified nested PCR products. Sequences were analyzed using a custom bioinformatics pipeline and were annotated with immunogenetic information using the Cloanalyst software package (https://www.bu.edu/computationalimmunology/research/software/) [31]. Identification of Ig subtypes and functional Ig heavy and light chains were first performed using human Ig sequence database as previously described [28] and recombinant monoclonal antibodies (mAbs) were generated based on this analysis. Subsequently the immunogenetic characteristics of the recombinant mAbs were reanalyzed using a rhesus Ig sequence database once it became available.

## Statistical analysis

Due to the small number of previously archived samples used in this study, the statistical analysis is mostly descriptive. Antibody binding data were analyzed by a non-parametric Mann-Whitney U test with p-value of $<0.05$ considered as significant. Correlation analysis was performed using Spearman's rank correlation test, with p-value of $<0.05$ considered as

significant. All statistical analyses were performed using GraphPad Prism version 9.0.0 for Windows, GraphPad Software, San Diego, CA, USA.

## Results

### TLR agonist adjuvants showed enhanced pediatric HIV Env antibody responses despite similar antigen-specific memory B cell frequencies in PBMC and tissues across vaccination groups

We have evaluated different vaccine regimens and immunization schedules (Fig 1, see details in methods) to optimize HIV Env-specific antibody responses in infant rhesus macaques [25, 26]. Previously, we observed that while the peak immunogenicity across different vaccine groups was comparable the magnitude and quality of vaccine-induced HIV Env-specific responses was enhanced by increasing the timing of vaccination interval from 3 to 6 weeks [25] and by using the TLR7/8 agonist adjuvant 3M-052-SE when compared to alum or to the TLR4 ligand glucopyranosyl lipid formulated in SE (GLA-SE) [26]. Since these analyses were conducted in two separate studies [25, 26], here we compare the responses at peak immunogenicity among all vaccine groups. Overall, infants vaccinated with 3M-052-SE-adjuvanted vaccine developed the highest magnitude of HIV Env C.1086 gp120-specific plasma IgG antibody concentrations across all four groups at peak immunogenicity (Fig 2A). Additionally, at peak immunogenicity, infant plasma antibody from all four groups developed varying levels of cross clade Env gp120 responses as well as broad heterologous epitope specificities and breadth (S2A and S2B Fig). Infant plasma IgG antibody from group 4 (3M-052-SE) also demonstrated higher avidity strengths against the tested antigens (HIV Env 1086d7gp120 K160N and gp70 ConC V3) compared to other immunization groups (S2C Fig).

### Single-cell flow cytometry sorting of antigen-specific memory B cells indicated low level of frequencies across all vaccination groups

To determine whether the different vaccine regimens induced distinct frequency of antigen-specific memory B cells, we characterized antigen-specific memory B cells from PBMC and tissues at selected time points (Fig 1). We were able to obtain Env-specific memory B cells (CD3-CD16-CD14-CD20+CD27+IgD-, double positive for HIV Env C.1086 gp120) from a total of 15 infants, including 3 of 5 infants in group 1 (HIV Env+adjuvant), 5 of 5 infants in group 2 (Co-administration), 3 of 5 infants in group 3 (Extended Interval), and from 4 of 4 infants in group 4 (3M-052-SE) (Table 1). The frequency of Env-specific memory B cells were low across all vaccine groups (0.06–1.72%) with 0.07–1.72% range in group 1, 0.06–0.97% in group 2, 0.08–0.49% in group 3, and 0.06–0.11% in group 4 (Table 1). The frequency of antigen-specific memory B cells were not correlated with the Env-specific IgG responses against the HIV Env 1086c gp120 K160N, although the small sample size could be a major limitation of the analysis (Fig 2B).

B cell receptor (BCR) repertoires analysis using the human Ig-gene sequence database led to identification of functional heavy- and light-chain pairs, which were used to produce a total of 39 monoclonal antibodies (mAbs). The human Ig-gene sequence database was utilized as the rhesus Ig-gene database was unavailable at the time when the initial analyses were performed [28]. The 39 mAbs came from 13 specimens including 3 specimens from group 1 (2 infants), 3 specimens from group 2 (2 infants), 4 specimens from group 3 (2 infants), and 3 specimens from group 4 (3 infants) for final B cell repertoire analyses (S1 Table). No tissue specimen was available for group 4 as these infants were part of a challenge study. Overall, the frequencies of Env-specific memory B cells in PBMCs and tissues did not differ between the

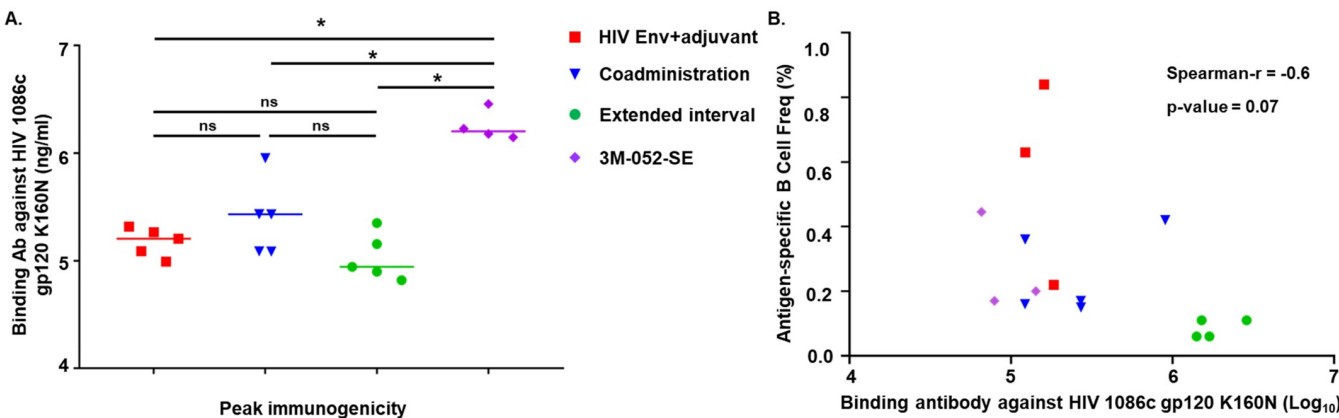

**Fig 2. Vaccine-induced Env binding IgG responses against HIV 1086c gp120 K160N does not correlate with frequency of antigen-specific memory B cells in infants.** (A) Binding antibody level was significantly higher in the 3M-052-SE group as measured by ELISA at peak immunogenicity. (B) Binding antibody levels at peak immunogenicity was not correlated with antigen-specific memory B cell frequencies in all vaccine groups. Statistical analyses were performed with GraphPad Prism, * denoted significant p-values of <0.05 by a non-parametric Mann-Whitney test. Correlation analysis was performed using Spearman's rank correlation test with GraphPad Prism, with p-value of <0.05 considered as significant.

vaccine groups. The percent frequency of antigen-specific memory B cells ranged from 0.06 to 1.72%, with the highest frequency observed in the PBMCs of 2 infants in group 1 (HIV Env+-adjuvant) (Table 1). Notably, despite higher magnitude and quality of Env-specific plasma antibody responses in group 3 (Extended Interval) compared to group 2 (Co-administration) [25], the frequencies of Env-specific memory B cells did not differ between these groups. Similarly, despite higher binding magnitude and avidity strength of Env-specific plasma antibody responses in group 4 (3M-052-SE) when compared to others, the frequencies of Env-specific memory B cells at least in the PBMCs of vaccinated infants in this group did not differ to the other groups. Altogether, these data suggest that the size of HIV Env vaccine-elicited memory B cell pool is not directly related to magnitude or quality of plasma Env-specific antibody responses in infant rhesus macaques.

## Epitope specificities and immunogenetics of vaccine-elicited Env-specific memory BCR repertoires did not differ across vaccination groups

We next characterized the immunogenetics and epitope specificities of the HIV Env vaccine-elicited mAbs. In our initial analysis, the identification of functional heavy- and light-chains and their immunogenetic characteristics was conducted using a previously validated bioinformatic method for rhesus BCR repertoire characterization using the human Ig sequence database [28, 30] (S1 Table). Notably, the majority of identified functional heavy- and light-chain Ig pairs were of IgG isotype (31 functional pairs) with IgA and IgM isotypes represented in 4 and 3 functional pairs, respectively (S3 Fig and S1 Table). Most mAbs were specific to the HIV Env V3 loop (14 functional Ig pairs) followed by V1V2 loop-specific mAbs (5 functional pairs). Only one functional pair targeting the CD4 binding site was identified, and we were unable to determine the epitope specificities of 18 functional heavy and light chains pairs. These data are consistent with our prior observations which demonstrated that most of the polyclonal plasma antibody response in these vaccinated infant macaques was directed against the HIV Env V3 region [25, 26].

Recent advances in genome sequencing and detailed characterization of rhesus Ig loci provided better understanding of allelic diversity in rhesus Ig genes [32–34]. Moreover, the

**Table 2. Immunogenetic characteristics of isolated envelope (Env)-reactive mAbs of Env-vaccinated infant monkeys based on rhesus macaque immunoglobulin database analysis.**

| Animal ID | Group | Tissue | IgH ID | $V_H$ gene | $D_H$ gene | $J_H$ gene | HC % SHM | HC CDR3 length | Ig Isotype | IgL ID | $V_L/V_\kappa$ gene | $J_L/J_\kappa$ gene | Specificity |
|---|---|---|---|---|---|---|---|---|---|---|---|---|---|
| 45521 | HIV Env+adjuvant | Spleen | H691207 | IGHV4-j*02 | IGHD3-9*01 | IGHJ5-2*01 | 3.82 | 23 | IgG | K690414 | IGKV1-n*01 | IGKJ2-1*01 | Undetermined |
| 45521 | HIV Env+adjuvant | Retropharyngeal LN | H691248 | IGHV4-j*02 | IGHD3-9*01 | IGHJ5-2*01 | 5.21 | 23 | IgA | K690428 | IGKV1-n*01 | IGKJ2-1*01 | Undetermined |
| 45521 | HIV Env+adjuvant | Retropharyngeal LN | H691255 | IGHV4-j*03 | IGHD3-9*01 | IGHJ5-2*01 | 4.51 | 23 | IgG | K690431 | IGKV1-n*01 | IGKJ2-1*01 | Undetermined |
| 45521 | HIV Env+adjuvant | Retropharyngeal LN | H691256 | IGHV4-j*02 | IGHD3-9*01 | IGHJ5-2*01 | 3.82 | 23 | IgG | K690432 | IGKV1-n*01 | IGKJ2-1*01 | Undetermined |
| 45522 | HIV Env+adjuvant | Spleen | H691308 | IGHV4-n*01 | IGHD6-24*01 | IGHJ4*01 | 7.90 | 13 | IgG | L690936 | IGLV1-e*01 | IGLJ3*01 | Undetermined |
| 45083 | Coadministration | Spleen | H691279 | IGHV4-f*02 | IGHD6-34*01 | IGHJ4*01 | 7.56 | 13 | IgM | L690918 | IGLV1-b*01 | IGLJ3*01 | V1V2 |
| 45083 | Coadministration | Mediastinal LN | H691285 | IGHV4-g*02 | IGHD3-26*01 | IGHJ4*01 | 4.17 | 17 | IgG | K690448 | IGKV1-a*01 | IGKJ2-1*01 | V1V2 |
| 45083 | Coadministration | Mediastinal LN | H691285 | IGHV4-g*02 | IGHD3-26*01 | IGHJ4*01 | 4.17 | 17 | IgG | L690922 | IGLV2-i*01 | IGLJ1*01 | V3 |
| 45083 | Coadministration | Mediastinal LN | H691289 | IGHV4-f*03 | IGHD5-5*02 | IGHJ4*01 | 5.15 | 13 | IgG | L690925 | IGLV2-a*01 | IGLJ1*01 | Undetermined |
| 45091 | Coadministration | Axillary LN | H691299 | IGHV3-g*03 | IGHD4-4*02 | IGHJ6*01 | 6.80 | 14 | IgG | L690932 | IGLV11-a*01 | IGLJ2*03 | Undetermined |
| 45091 | Coadministration | Axillary LN | H691306 | IGHV4-j*02 | IGHD4-22*01 | IGHJ4*01 | 5.15 | 17 | IgG | K690454 | IGKV1-e*05 | IGKJ4-1*01 | V3 |
| 45091 | Coadministration | PBMC | H691004 | IGHV3-a1*01 | IGHD3-3*01 | IGHJ4*01 | 7.99 | 19 | IgG | L690726 | IGLV3-c*02 | IGLJ2*03 | CD4 binding site |
| 45435 | Extended Interval | Mediastinal LN | H691315 | IGHV4-f*03 | IGHD3-21*01 | IGHJ3*01 | 7.22 | 18 | IgG | K690458 | IGKV1-f*04 | IGKJ4-1*01 | V1V2 |
| 45435 | Extended Interval | Mediastinal LN | H691312 | IGHV4-j*02 | IGHD1-39*01 | IGHJ5-1*01 | 5.56 | 16 | IgG | K690456 | IGKV1-n*01 | IGKJ2-1*01 | CD4 binding site |
| 45435 | Extended Interval | Mediastinal LN | H691321 | IGHV3-a1*01 | IGHD3-14*01 | IGHJ6*01 | 2.43 | 17 | IgG | K690463 | IGKV1-r*01 | IGKJ2-1*01 | V1V2 |
| 45435 | Extended Interval | Mediastinal LN | H691322 | IGHV3-r*02 | IGHD6-29*01 | IGHJ6*01 | 2.43 | 12 | IgG | K690464 | IGKV3-c*01 | IGKJ1-1*01 | V1V2 |
| 45441 | Extended Interval | Spleen | H691341 | IGHV3-y*03 | IGHD4-22*01 | IGHJ6*01 | 4.86 | 0 | IgG | K690472 | IGKV1-a*01 | IGKJ1-1*01 | Undetermined |
| 45441 | Extended Interval | Spleen | H691346 | IGHV4-j*02 | IGHD3-21*01 | IGHJ4*01 | 13.75 | 0 | IgG | L690948 | IGLV1-e*01 | IGLJ7*01 | CD4 binding site |
| 45441 | Extended Interval | Spleen | H691343 | IGHV3-y*03 | IGHD2-25*01 | IGHJ1*01 | 5.21 | 15 | IgG | L690946 | IGLV2-j*16 | IGLJ6*01 | Undetermined |
| 45441 | Extended Interval | Submental LN | H691216 | IGHV4-j*02 | IGHD3-3*01 | IGHJ6*01 | 1.39 | 18 | IgA | K690417 | IGKV2-d*03 | IGKJ1-1*01 | Undetermined |
| 45851 | 3M-052-SE | PBMC | H680141 | IGHV3-l*02 | IGHD5-23*01 | IGHJ5-2*01 | 0.00 | 11 | IgG | K680049 | IGKV2-r*01 | IGKJ1-1*01 | Undetermined |
| 45851 | 3M-052-SE | PBMC | H680144 | IGHV4-m*02 | IGHD1-A*01 | IGHJ4*01 | 2.38 | 13 | IgG | K680048 | IGKV2-g*01 | IGKJ1-1*01 | Undetermined |
| 45851 | 3M-052-SE | PBMC | H680146 | IGHV4-f*02 | IGHD6-24*01 | IGHJ4*01 | 4.47 | 10 | IgD | L680105 | IGLV5-b*01 | IGLJ1*01 | V3 |
| 45851 | 3M-052-SE | PBMC | H680147 | IGHV5-b*02 | IGHD7-A*01 | IGHJ6*01 | 2.43 | 13 | IgG | L680103 | IGLV1-d*01 | IGLJ2*03 | V3 |
| 45851 | 3M-052-SE | PBMC | H680149 | IGHV4-f*02 | IGHD6-24*01 | IGHJ4*01 | 5.15 | 10 | IgE | L680109 | IGLV6-c*01 | IGLJ2*03 | V3 |
| 45851 | 3M-052-SE | PBMC | H680150 | IGHV4-m*02 | IGHD1-A*01 | IGHJ4*01 | 2.04 | 13 | IgG | L680108 | IGLV5-b*01 | IGLJ1*01 | V3 |

A total of 26 pairs of potentially Env-reactive mAbs were isolated from the four vaccination groups across several anatomic compartments. Frequency of gene usage, percent somatic hypermutation, and complementarity-region 3 (CDR3) length are displayed for the heavy and light chains for each mAb along with the isotype and epitope specificity.

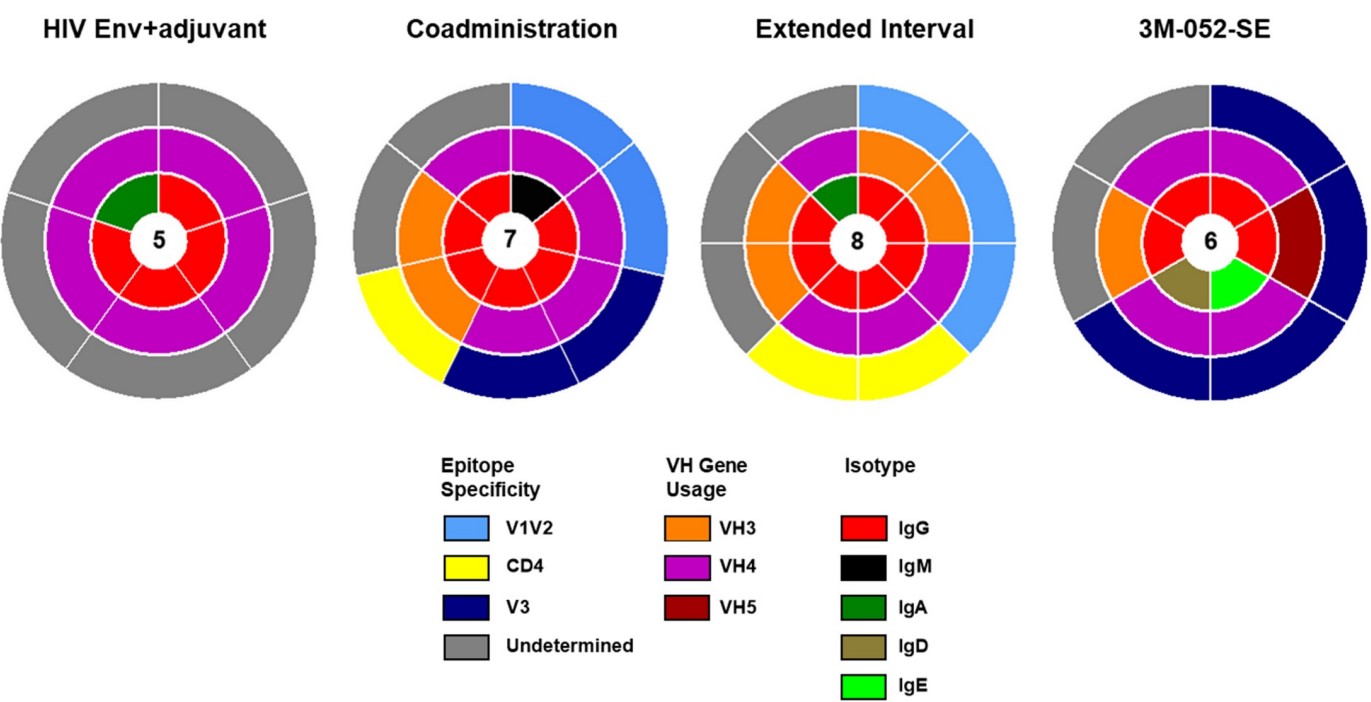

**Fig 3. Analyses of epitope specificity and immunogenetic characteristics of the Env-specific functional heavy- and light-chains of 26 vaccine-elicited mAbs in infants using rhesus Ig-gene database.** Reanalysis using newly developed software based on rhesus macaque Ig sequences confirmed that 26 of 39 heavy- and light-chain pairs previously identified using human Ig sequences were functional. Overall, the epitope specificities, VH gene family usage, and isotype distribution were similar across different vaccina groups. Epitope specificity, VH gene family usage, and isotype distribution of identified functional heavy and light chains are displayed in concentric circles. The number of mAbs per group is displayed in the center.

availability of rhesus Ig gene libraries [32] provided the opportunity to reanalyze the 39 functional Ig pairs that we previously identified using the human Ig sequence database. This secondary analysis confirmed that 26 of 39 Ig pairs were indeed functional (Table 2) with 5 Ig pairs in group 1 (HIV Env+adjuvant), 7 Ig pairs in group 2 (Co-administration), 8 Ig pairs in group 3 (Extended Interval), and 6 Ig pairs in group 4 (3M-052-SE) (Fig 3). Similar to earlier analysis using the human Ig sequence database, the majority of identified functional heavy- and light-chain Ig pairs are of IgG isotype (21 functional pairs). Based on the rhesus Ig sequence database, the variable heavy chain (VH) gene usage was largely restricted to the VH3 and VH4 gene families, across all the vaccination groups, and there was no apparent difference in VH usage in the 3M-052-SE group as compared to the other vaccination groups (Table 2). The majority of the Ig pairs confirmed as functional in the secondary analysis using rhesus Ig-gene database were against undefined epitopes (12 Ig pairs), whereas 6 pairs were specific to the V3 loop and 5 pairs targeted the V1V2 loop. this suggests that an important proportion of the vaccine-elicited mAbs were against non-linear conformational epitopes on the HIV Env.

### Frequency of somatic hypermutation and heavy chain complementarity-determining region 3 (HCDR3) length in HIV Env-specific Ig pairs did not differ across vaccination groups

We also sought to define whether increased somatic hypermutation and affinity maturation in Env-specific Ig pairs could potentially contributed to distinct plasma antibody responses

induced by different pediatric HIV vaccine regimens [25, 26]. Based on the preliminary Ig sequence analysis using the human Ig gene database, the frequency of somatic hypermutation (SHM) was comparable across the vaccination groups (S4 Fig) with range of SMH of 5.31–10.03% in group 1 (HIV Env+adjuvant), 4.52–10.15% in group 2 (Co-administration), 6.05–11.08% in group 3 (Extended Interval), and 4.15–7.03% in group 4 (3M-052-SE) (S1 Table). Additionally, the HCDR3 of Env-reactive functional pairs were comparable across all vaccination groups, with HCDR3 length range of 10–23 aa. The highest median HCDR3 length was observed in the HIV Env+adjuvant group (median length of 23 aa) and the lowest was observed in the 3M-052-SE group (median length of 13 aa). The HCDR3 median length in group 2 (Co-administration) and 3 (Extended Interval) was 14 aa and 17 aa, respectively.

Comparable SHM frequency across the groups was confirmed with the reanalysis of the human Ig sequence database using the rhesus Ig sequence database (Fig 4 and Table 2), albeit mutation rates were slightly lower compared to the initial analysis using the human Ig sequence database. The range of SHM was 3.82–7.9% in group 1 (HIV Env+adjuvant), 4.17–7.99% in the group 2 (Co-administration), 1.39–13.75% in group 3 (Extended Interval), and 0–5.15% in group 4 (3M-052-SE) (Table 2). The HCDR3 length of Env-reactive functional pairs was also comparable across all vaccination groups. Similar to human Ig sequence, the highest median HCDR3 region was observed in group 1 (HIV Env+adjuvant) and the lowest median HCDR3 region was observed in group 4 (3M-052-SE). Interestingly, the median HCDR3 length in group 2 (Co-administration) was lower based on human Ig sequence database (median HCDR3 length of 14 aa) than the rhesus Ig sequence (median HCDR3 length of 17 aa). Meanwhile, the median HCDR3 length for group 3 (Extended Interval) was lower with the rhesus IgG sequence (median HCDR3 length of 15 aa) than in the human Ig sequence (median HCDR3 length of 17 aa). These findings highlight the limitations of using human database to analyze the BCR repertoire in rhesus.

Altogether, our data suggest that the magnitude and quality of vaccine-elicited plasma Env-specific antibody responses administered with different adjuvants are not related to the size of the antigen-specific memory B cell pool or to the immunogenetics characteristics of the vaccine-elicited Ig pairs.

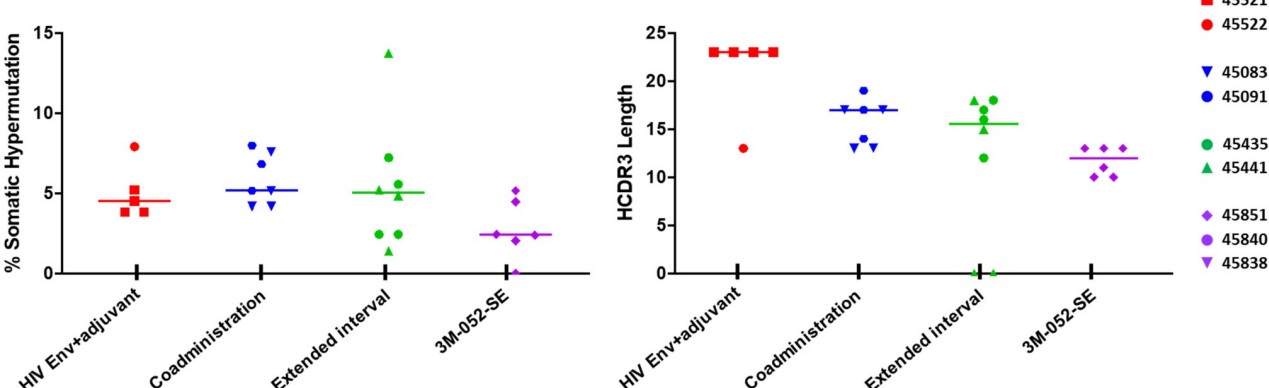

**Fig 4. Frequency of somatic hypermutation and heavy chain complementarity-determining region 3 (HCDR3) length of vaccine-elicited Env-reactive functional heavy- and light-chains identified using rhesus Ig sequence database.** (A) Analysis of percent somatic hypermutation frequency and HCDR3 lengths for Env-reactive heavy and light chains pairs (39 mAb pairs) from infant antigen-specific B cells based on human immunoglobulin (Ig) sequence database. (B) Analysis of percent somatic hypermutation frequency and HCDR3 lengths for Env-reactive heavy- and light-chains functional pairs (26 mAb pairs) from vaccinated infant antigen-specific memory B cells using rhesus Ig-gene database. Horizontal lines indicated median values of individual groups. Corresponding functional heavy and light chains isolated from individual infants are denoted by symbols.

## Discussion

Based on previous observations that different pediatric HIV vaccine regimens induced distinct plasma antibody responses [25, 26], the goal of this study was to investigate whether these responses were driven by distinct memory BCR repertoire characteristics including distinct Ig gene usage, rates of SHM and HCDR3 lengths. We utilized samples from 2 completed immunization studies [25, 26], in which newborn infant rhesus macaques were immunized with four distinct vaccine regimens (Fig 1). Our results indicate that although the different vaccine regimens induced distinct plasma antibody responses, there were no significant differences in the B cell repertoires.

The development of a safe and effective pediatric HIV-1 vaccine to eliminate postnatal infant HIV-1 infections will probably require the use of novel adjuvants such as Toll-like receptor (TLR) agonists. Indeed, recent studies have demonstrated that TLR7/8 adjuvantation can enhance vaccine responses in early life [35–37]. Notably, our group previously reported that an HIV vaccine adjuvanted with 3M-052-SE induces superior plasma antibody levels than other adjuvanted HIV regimens in infant rhesus macaques [26]. We showed that antibody levels against autologous and heterologous envelope proteins, and observed that overall, the magnitude of the vaccine-elicited antibody response was higher in the 3M-052-SE group (Fig 2A and S2A–S2C Fig). However, the impact of these different pediatric HIV Env vaccine regimens on the developing infant antigen-specific B cell BCR repertoires is still unclear.

Previous study investigating the BCR repertoire in HIV immunized adult monkeys have reported a preferential usage of the VH4 and VH3 families [28]. Our initial analysis of the BCR repertoire in the vaccinated infant rhesus macaques was conducted using the same human Ig sequence database and bioinformatics methods, in which we similarly observed that 69% of the infant Env-specific functional heavy and light chains in this study used VH4 genes and 26% use VH3 genes (S1 Table). However, we found that the Env-specific functional heavy and light chains in infant rhesus macaques were slightly shorter and had lower mutation rates than in adults. For example, in adult rhesus macaques immunized with a pox prime/protein boost regimen the median HCDR3 length after the fifth immunization was 16 aa and the SHM rate was 9.3%. Meanwhile, in our study the overall median HCDR3 length across all vaccination groups was 14 aa and the SMH rate was 6.3%. The lower levels of SHM rate in infant rhesus macaques as compared to adult rhesus macaques is in accordance with the observation that SHM rate increases with age. Moreover, broadly neutralizing antibodies (bNAbs) isolated from HIV-infected pediatric patients (1-year post-infection) also have lower levels of SHM rate than HIV-infected adult bNAbs directed against the same epitopes [38].

Due to their close genetic similarity to humans, rhesus macaques are valuable animal model for studies of infectious diseases including the understanding of vaccine-elicited immune responses. However, despite their wide usage as a human surrogate model system, many aspects of the rhesus macaque immune system are still under characterized and poorly annotated. Recent advances in high-throughput NGS and specialized computational methods provide tools to the scientific community to compare rhesus repertoires of heavy and light-chains to humans in order to better understand how they may perform as a model system for B- and T-cell mediated immunity in humans [39–41]. Characterization of human and rhesus BCR repertoires showed that the frequency of V- and J-gene segment usage and HCDR3 lengths between human and rhesus were in concordance with one another [39, 42]. However, comparative analyses of different Ig subtypes (IgM, IgG, IgK and IgL) sequences revealed significant differences in the overall BCR repertoires. Rhesus macaques have higher diversity of BCR

repertoires with different family gene usage and slight difference in the frequencies of HCDR3 lengths within the IgM$^+$ BCR repertoires, likely due to gene family usage in the class-switched (IgG) compartment [39]. However, importantly, the low abundance of long CDRH3s in rhesus IgM$^+$ B cells did not impede their expansion into the IgG$^+$ B cells, and in rhesus IgG$^+$ B cell frequency was comparable to human IgG$^+$ B cells. Thus, given the complexity of gene recombination, high diversity in rhesus BCR repertoires, and the close genetic relationship between rhesus and humans, it is likely that rhesus B-cell compartment recapitulates its human counterpart and is poised to respond to antigen in similar manners.

Historically, a limitation of BCR repertoire analysis in the nonhuman primate (NHP) model has been the lack of rhesus macaque Ig sequence database. Recent development of bioinformatics tools that enable comparison of Ig sequences from immunized animals to a NHP reference database allows for a more accurate characterization of the BCR repertoire in response to vaccination or infection. We found some differences in BCR repertoire characteristics including VH usage, SMH rate, and Ig subclass/isotypes when the same Ig pairs were analyzed using human Ig and rhesus Ig sequence databases (Figs 3 and 4, Table 2, S3 and S4 Figs, S1 Table). Notably only 26 of 39 pairs identified as functional based on the human Ig sequence database were confirmed to be functional using the rhesus Ig sequence database. This could be due to high sequence homology using the rhesus Ig sequence database compared to human Ig sequence database. These findings highlight the importance of species-specific database for comprehensive understanding of the BCR repertoire and antibody maturation in response to vaccinations and/or infections using the rhesus macaque model.

Interestingly, the observed SHM rate in infants across all vaccinations groups in this study (4.7%) is only slightly higher than the observed SHM rate in human HIV vaccine trials such as GSK PRO HIV-002 (3.8%) and the RV144 trial (2.4%). HIV-1-infected infants have been shown to develop neutralization breadth earlier than HIV-infected adults. Plasma antibody responses in HIV-infected infants neutralized a panel of diverse HIV-1 viruses, including more difficult to neutralize cross clade variants [43]. Additionally, these responses were observed as early as 1 to 2 years post-infection. Furthermore, bNAbs isolated from pediatric HIV cases appeared to have lower levels of SHM when compared to bNAbs isolated from HIV-infected adults [38]. Altogether, these data suggest that induction of HIV-1-specific plasma antibody neutralization can be achieved in children without prolonged extensive SHM and affinity maturation.

## Conclusion

Our results suggest that the high plasma antibody magnitude and functionality achieved with 3M-052-SE adjuvantation is not accompanied by distinct B cell repertoire characteristics. Nevertheless, our study has several limitations including, (1) small sample size, (2) distinct vaccine regimens and timing used, and (3) the low number of functional heavy and light chain pairs identified from the different vaccination groups. Nevertheless, this study provides novel insights into (1) the B cell repertoires in vaccinated infant rhesus macaques, which has not been analyzed, and (2) significant increase of antibody magnitude observed in our previous studies in the 3M-052-SE group was not related to distinct immunogenetics characteristics of vaccine-induced antibodies. Further investigation of the mechanism by which the 3M-052-SE adjuvant leads to enhanced immune responses in the setting of the developing early life immune system are warranted. Additionally, it will be important to evaluate whether the enhanced immunogenicity of HIV vaccines with 3M-052-SE adjuvantation in infants is associated with protection from oral virus exposure.

## Supporting information

**S1 Fig. A-J**. Representative flow cytometry analyses for single cell HIV Env-specific memory B cell sorting from PBMC and tissues of infants by vaccination groups. Percentage of memory B cells (CD20+CD27+IgD-), which were reactive to HIV 1086c gp120 K160N tagged with two colors-BV421 and AF647 (double positive cells) indicated in the upper right quadrant. Gates were drawn based on isotype controls and fluorescent minus one (FMO) controls. (PDF)

**S2 Fig. Characterization of vaccination-induced plasma envelope IgG responses in infant rhesus macaques.** (A) Cross clade HIV gp120-specific IgG responses at peak immunogenicity by luminex assay (BAMA). (B) Variable and conserved epitopes-specific IgG responses at peak immunogenicity by BAMA. (C) The 3M-052 group antibody responses had higher avidity strength against most of the tested antigens. Avidity 1/k off (the inverse of the dissociation rate) was plotted as a measure of the strength of binding and avidity scores which take into consideration the magnitude are also shown. Statistical analyses were performed with GraphPad Prism, * denoted significant p-values of $<0.05$ by a non-parametric Mann-Whitney test. (PDF)

**S3 Fig. Analyses of epitope specificity and immunogenetic characteristics of the Env-specific functional heavy- and light-chains of 39 vaccine-elicited mAbs in infants using human Ig-gene database.** Initial analysis with human immunoglobulin (Ig) database indicated a total of 39 heavy- and light-chain pairs isolated from antigen-specific memory B cells across different vaccine groups. Epitope specificity, VH gene family usage, and isotype distribution of identified functional heavy- and light-chain pairs were similar across vaccine groups. Epitope specificity, VH gene family usage, and isotype distribution of identified functional heavy and light chains are displayed in concentric circles. The number of mAbs per group is displayed in the center. (PDF)

**S4 Fig. Frequency of somatic hypermutation and heavy chain complementarity-determining region 3 (HCDR3) length of vaccine-elicited Env-reactive functional heavy- and light-chains identified using rhesus Ig sequence database.** Analysis of percent somatic hypermutation frequency and HCDR3 lengths for Env-reactive heavy and light chains pairs (39 mAb pairs) from infant antigen-specific B cells based on human immunoglobulin (Ig) sequence database. Horizontal lines indicated median values of individual groups. Corresponding functional heavy and light chains isolated from individual infants are denoted by symbols. (PDF)

**S1 Table. Immunogenetic characteristics of isolated envelope (Env)-reactive mAbs of Env-vaccinated infant monkeys based on human immunoglobulin database analysis.** A total of 39 pairs of potentially Env-reactive mAbs were isolated from the four vaccination groups across several anatomic compartments. Frequency of gene usage, percent somatic hypermutation, and complementarity-region 3 (CDR3) length are displayed for the heavy and light chains for each mAb along with the isotype and epitope specificity. (PDF)

**S1 Data.**
(PDF)

## Acknowledgments

We would like to thank Carolyn Weinbaum for her contribution in animal study and sample coordination. Flow cytometry and single cell sorting was performed at the Duke Human Vaccine Institute (DHVI) Flow Cytometry Facility, Duke University, Durham, NC.

## Author Contributions

**Conceptualization:** Kristina De Paris, Sallie R. Permar, M. Anthony Moody, Genevieve G. Fouda.

**Data curation:** Stella J. Berendam, Shuk Hang Li.

**Formal analysis:** Stella J. Berendam, Papa K. Morgan-Asiedu, Shuk Hang Li, Michael Hudgens, Kevin Wiehe.

**Funding acquisition:** Kristina De Paris, Sallie R. Permar.

**Investigation:** Papa K. Morgan-Asiedu, Riley J. Mangan, Holly Heimsath, Alan D. Curtis, II, Joshua A. Eudailey, Bonnie Phillips, Hannah L. Itell, Erika Kunz.

**Methodology:** Kan Luo, Alan D. Curtis, II, Michael Hudgens, Kenneth Cronin, Kevin Wiehe, S. Munir Alam.

**Project administration:** Koen K. A. Van Rompay.

**Resources:** Christopher B. Fox, Mark A. Tomai.

**Supervision:** Koen K. A. Van Rompay, Kristina De Paris, Sallie R. Permar, M. Anthony Moody, Genevieve G. Fouda.

**Visualization:** Stella J. Berendam.

**Writing – original draft:** Stella J. Berendam.

**Writing – review & editing:** Kevin Wiehe, Koen K. A. Van Rompay, Kristina De Paris, Sallie R. Permar, M. Anthony Moody, Genevieve G. Fouda.

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
