## [Decision Letter · Decision Letter 0]

27 Sep 2021

PONE-D-21-26264Different adjuvanted pediatric HIV envelope vaccines induced distinct plasma antibody responses despite similar B cell receptor repertoires in infant rhesus macaques.PLOS ONE

Dear Dr. Fouda,

Thank you for submitting your manuscript to PLOS ONE. After careful consideration, we feel that it has merit but does not fully meet PLOS ONE’s publication criteria as it currently stands. Therefore, we invite you to submit a revised version of the manuscript that addresses the points raised during the review process.

We look forward to receiving your revised manuscript.

Kind regards,

Mrinmoy Sanyal, PhD

Academic Editor

PLOS ONE

Journal Requirements:

“This work was supported by grant P01 AI117915 from the (National Institute of Allergy and Infectious Diseases (NIAID) to KDP and SRP, and P51OD011107 (Office of Research Infrastructure Program, Office of The Director, NIH) to CNPRC. SJB is supported by the Interdisciplinary Research and Training Program in AIDS (5T32AI007392-31), Duke University.”

“This work was supported by grant P01 AI117915 from the (National Institute of Allergy and Infectious Diseases (NIAID) to KDP and SRP, and P51OD011107 (Office of Research Infrastructure Program, Office of The Director, NIH) to CNPRC. SJB is supported by the Interdisciplinary Research and Training Program in AIDS (T32), Duke University.”

Reviewers' comments:

Reviewer's Responses to Questions

**Comments to the Author**

1. Is the manuscript technically sound, and do the data support the conclusions?

Reviewer #1: Partly

Reviewer #2: Yes

2. Has the statistical analysis been performed appropriately and rigorously? 

Reviewer #1: No

Reviewer #2: I Don't Know

3. Have the authors made all data underlying the findings in their manuscript fully available?

Reviewer #1: Yes

Reviewer #2: Yes

4. Is the manuscript presented in an intelligible fashion and written in standard English?

Reviewer #1: Yes

Reviewer #2: Yes

5. Review Comments to the Author

Reviewer #1: Berendam et al. analyzed Env-specific memory B cell responses in infant rhesus macques immunized with (i) HIV Env protein with a squalene adjuvant, (ii) MVA-HIV and Env protein co-administered using a 3-week interval, (iii) MVA-HIV prime/protein boost with an extended 6–61-week interval between immunizations, or (iv) with HIV Env administered with 3M-052-SE adjuvant. Using banked samples from previous studies, the authors showed that there is no difference in the magnitude of Env-specific memory B cell frequencies, somatic hypermutation or HCDR3 length. Additionally, they sorted single B cells and produced 39 monoclonal antibodies. A simple and good study; however, “n” is too small to identify significant effects.

The manuscript could be improved by addressing the following.

1. Fig. 1 suggests that memory B cells were sorted 2 w post three immunizations in groups 1 and 2, but 9 weeks post three immunizations in group 4. However, supplementary fig. 1 suggest the PBMC sorts were 10-12 weeks. Could the authors clarify?

2. If fig. 1 is correct, then the time points are not comparable. Can the authors use a time point common between the groups for sorting?

3. A technical point. In supplementary fig. 1b, selection of CD20 includes so many cells that are double-positive for CD3 and CD20. It looks like a technical problem. Not sure how much does that affect the final outcome given that we are looking at really minor populations.

4. Can the authors show the final plot (Probe 1 vs. probe 2) for all samples in a supplementary figure, or even a main figure? Most papers do that and it’s not hard given the small number of samples.

5. Fig. 2 protein only, does it not have an adjuvant? It’s misleading to say protein only.

Reviewer #2: Berendam and colleagues tested if pediatric HIV envelope vaccine regimens with different adjuvants induced distinct antigen-specific memory B cell repertoires, and if specific Ig immunogenetic characteristics are associated with a higher magnitude of plasma antibody responses. Intriguingly, the study showed that the frequency of memory B cells was similar across regimens and uncorrelated with the level of plasma antibody. This is an interesting study that will be of interest to the field. The most important data are in supplemental figures, which should be promoted to main figures.

Major points:

1. The data in Figs. S1-S2 are interesting and central to the paper, so they should be included as main figures. The differences in bAb concentration in Fig. S1 and memory B cells in S2 are the core findings of the manuscript.

2. Analysis of the sequenced antibodies using the human Ig-gene database is irrelevant, especially because an analysis using the macaque database is also presented. Table 1 and Fig. 2A should both be removed. Now that the macaque databases are available, I do not think anybody will perform analyses based on the human database--so readers will not be interested. In my opinion the entire preliminary analysis using the human Ig-gene database can be removed from the paper.

3. Regarding statistics, it is stated that "There was no association between vaccine-elicited antigen-specific memory B cell frequencies and plasma antibody titer or avidity." However, as far as I can tell, no details whatsoever are given to explain the statistical test performed in which no association was uncovered. Looking at the supplemental figures, in fact, it appears possible that there was an /inverse/ association between Bmem frequencies and bAb concentrations.

6. PLOS authors have the option to publish the peer review history of their article (what does this mean?). If published, this will include your full peer review and any attached files.

Reviewer #1: No

Reviewer #2: No

---

## [Author Response · Author response to Decision Letter 0]

16 Nov 2021

PONE-D-21-26264 Different adjuvanted pediatric HIV envelope vaccines induced distinct plasma antibody responses despite similar B cell receptor repertoires in infant rhesus macaques.

We thank the reviewers for their insightful comments that led to valuable improvements of our manuscript. We have incorporated all suggestions made by the reviewers in the marked-up version of the manuscript, which highlights changes made to the original version (document named as Revised Manuscript with Track Changes). Additionally, an unmarked version of our revised paper without tracked changes was also uploaded in the online resubmission (document named Manuscript). We also provided new supplemental data (document named as Supplemental Data) that addressed the reviewers’ suggestions by including representative flow sorting analyses of different animals and tissue types from all vaccine groups. Finally, please see below, in blue font, our point-by-point response to the comments/concerns/suggestions made by the reviewers. 

Reviewer #1: 

Berendam et al. analyzed Env-specific memory B cell responses in infant rhesus macaques immunized with (i) HIV Env protein with a squalene adjuvant, (ii) MVA-HIV and Env protein co-administered using a 3-week interval, (iii) MVA-HIV prime/protein boost with an extended 6–61-week interval between immunizations, or (iv) with HIV Env administered with 3M-052-SE adjuvant. Using banked samples from previous studies, the authors showed that there is no difference in the magnitude of Env-specific memory B cell frequencies, somatic hypermutation or HCDR3 length. Additionally, they sorted single B cells and produced 39 monoclonal antibodies. A simple and good study; however, “n” is too small to identify significant effects.

Response to Reviewer: As mentioned in our manuscript and by the reviewer, the small sample size in the study was due to avaibility of archived samples from 2 prior completed studies. We acknowledge that the small sample size in the study is a limitation but we feel that the findings are still relevant and informative to the field given the robust antibody responses previously reported in the 3M-052-SE group in comparison to other vaccine groups. 

The manuscript could be improved by addressing the following.

1. Fig. 1 suggests that memory B cells were sorted 2 w post three immunizations in groups 1 and 2, but 9 weeks post three immunizations in group 4. However, supplementary fig. 1 suggest the PBMC sorts were 10-12 weeks. Could the authors clarify? If fig. 1 is correct, then the time points are not comparable. Can the authors use a time point common between the groups for sorting? – We have adjusted the figure to reflect the time points used in the current manuscript instead of the original animal study schedules from each completed study (see Fig 1 in resubmission). To clarify further, PBMC sorts were performed at week 10-12 for group 1 (HIV Env+adjuvant), group 2 (Co-administration) and group 4 (3M-052.SE). No PBMC samples were available for sorting in group 3 (Extended Interval) at both week 10 and 12. B cells were also sorted from tissues at the necropsy time point for group 1, 2 and 3. Necropsy were performed at between 3 to 13 weeks after the last immunization. We acknowledge that comparable sampling across the groups would have been ideal, but we would like to point out that this is the first study looking at the B cell repertoire in infant rhesus macaques following HIV vaccination. Thus, although these samples of convenience are aligned to the original study designs, they provide novel information to the field.

2. A technical point. In supplementary fig. 1b, selection of CD20 includes so many cells that are double-positive for CD3 and CD20. It looks like a technical problem. Not sure how much does that affect the final outcome given that we are looking at really minor populations. – Given that HIV Env-specific memory B cells are relatively rare, we decided to “cast” a wider net in CD20+ gating but increased the stringency in gating what we call the “double positive” populations (upper right quadrant in Supplemental Figures 1A-J). These double positive populations were determined using both florescent minus one (FMO) controls as well as isotype antibody controls and unconjugated antigens. 

4. Can the authors show the final plot (Probe 1 vs. probe 2) for all samples in a supplementary figure, or even a main figure? Most papers do that and it’s not hard given the small number of samples. – We have included representative flow cytometry analyses plots for infants from different vaccination groups and sample types (see Supplemental Figure 1A-J). Additionally, we added the sorting details in Table 1, which include the number of sorting experiments performed for individual infants based on archived samples availability. 

5. Fig. 2 protein only, does it not have an adjuvant? It’s misleading to say protein only. – corrected, revised to HIV Env+adjuvant in all figures – We have changed the nomenclature for group 1 to HIV Env+adjuvant in the main text and figures to clarify the vaccine regimen.

Reviewer #2: Berendam and colleagues tested if pediatric HIV envelope vaccine regimens with different adjuvants induced distinct antigen-specific memory B cell repertoires, and if specific Ig immunogenetic characteristics are associated with a higher magnitude of plasma antibody responses. Intriguingly, the study showed that the frequency of memory B cells was similar across regimens and uncorrelated with the level of plasma antibody. This is an interesting study that will be of interest to the field. The most important data are in supplemental figures, which should be promoted to main figures.

Major points:

1. The data in Figs. S1-S2 are interesting and central to the paper, so they should be included as main figures. The differences in bAb concentration in Fig. S1 and memory B cells in S2 are the core findings of the manuscript. – We have moved the binding data against HIV Env 1086c gp120 (previously Sup Fig 1A) to the main text (see Fig 2A). We also added the B cell frequency (Sup Fig 2A) to the main text as a table (see Table 1).

2. Analysis of the sequenced antibodies using the human Ig-gene database is irrelevant, especially because an analysis using the macaque database is also presented. Table 1 and Fig. 2A should both be removed. Now that the macaque databases are available, I do not think anybody will perform analyses based on the human database--so readers will not be interested. In my opinion the entire preliminary analysis using the human Ig-gene database can be removed from the paper. – We appreciate the reviewer’s comment on the relevance of the human Ig data and agree that the analysis with the rhesus Ig gene database is the most relevant. However, we feel that it is essential for us to include the human Ig-gene analysis in the manuscript as the primary analyses of the sequences were performed using the human Ig-gene database as the rhesus Ig-gene database was not available yet. The secondary analyses were conducted when the rhesus Ig-gene database became available in the course of the study. Therefore, removing the analyses using the human Ig-gene database could potentially create a bias. Additionally, we feel that it is important to contrast the differences and similarities of B cell repertoires when analyzed using the two species specific database to highlight the potential effect of species-specific database for B cell repertoire studies. Furthermore, the initial data obtained with the human Ig database allowed us to compare the infant B cell repertoire in our study to published adult rhesus (ref 28, PMID: 27942585). However, we agree with the reviewer that the analyses using the human Ig are not the most relevant to the manuscript and we have moved the corresponding figures to the supplemental (see Sup Fig 3).

3. Regarding statistics, it is stated that "There was no association between vaccine-elicited antigen-specific memory B cell frequencies and plasma antibody titer or avidity." However, as far as I can tell, no details whatsoever are given to explain the statistical test performed in which no association was uncovered. Looking at the supplemental figures, in fact, it appears possible that there was an /inverse/ association between Bmem frequencies and bAb concentrations. – We thank the reviewer for the suggestion and have included the statistical analyses write up in the revised manuscript, which was not previously included in the first submission. The statistical analyses were limited due to low sample numbers in the study. Binding antibody analyses were performed using a non-parametric Mann-Whitney U Test in GraphPad Prism (version 9) in Fig 2A. However, we feel that the data from the study are still interesting and warrants future investigation with higher sample size. We also appreciate the suggestion to correlate the antigen-specific memory B cell frequencies and binding antibody levels against HIV Env 1086c gp120 and have performed correlation analysis using Spearman’s rank correlation test in GraphPad Prism (version 9) in Fig 2B. We observed no significant correlation between the two, however, we acknowledge that this could be due to small sample size.

---

## [Decision Letter · Decision Letter 1]

10 Dec 2021

Different adjuvanted pediatric HIV envelope vaccines induced distinct plasma antibody responses despite similar B cell receptor repertoires in infant rhesus macaques.

PONE-D-21-26264R1

Dear Dr. Fouda,

We’re pleased to inform you that your manuscript has been judged scientifically suitable for publication and will be formally accepted for publication once it meets all outstanding technical requirements.

Kind regards,

Mrinmoy Sanyal, PhD

Academic Editor

PLOS ONE

======

Reviewers' comments:

Reviewer's Responses to Questions

**Comments to the Author**

1. If the authors have adequately addressed your comments raised in a previous round of review and you feel that this manuscript is now acceptable for publication, you may indicate that here to bypass the “Comments to the Author” section, enter your conflict of interest statement in the “Confidential to Editor” section, and submit your "Accept" recommendation.

Reviewer #1: All comments have been addressed

Reviewer #2: All comments have been addressed

2. Is the manuscript technically sound, and do the data support the conclusions?

Reviewer #1: Yes

Reviewer #2: Yes

3. Has the statistical analysis been performed appropriately and rigorously? 

Reviewer #1: N/A

Reviewer #2: Yes

4. Have the authors made all data underlying the findings in their manuscript fully available?

Reviewer #1: Yes

Reviewer #2: Yes

5. Is the manuscript presented in an intelligible fashion and written in standard English?

Reviewer #1: Yes

Reviewer #2: Yes

6. Review Comments to the Author

Reviewer #1: (No Response)

Reviewer #2: Berendam and colleagues tested if pediatric HIV envelope vaccine regimens with different adjuvants induced distinct antigen-specific memory B cell repertoires, and if specific Ig immunogenetic characteristics are associated with a higher magnitude of plasma antibody responses. Intriguingly, the study showed that the frequency of memory B cells was similar across regimens and uncorrelated with the level of plasma antibody. The manuscript has been revised and greatly improved. The study will be of great interest to the field.

Minor points:

1. I think the heading "Single-cell flow cytometry sorting of antigen-specific memory B cells indicated low level of frequencies across all vaccination groups" is slightly incorrect. It could read instead, "Single-cell flow cytometry sorting of antigen-specific memory B cells indicated low frequencies across all vaccination groups".

7. PLOS authors have the option to publish the peer review history of their article (what does this mean?). If published, this will include your full peer review and any attached files.

Reviewer #1: No

Reviewer #2: No

---

## [Editor Report · Acceptance letter]

16 Dec 2021

PONE-D-21-26264R1 

Different adjuvanted pediatric HIV envelope vaccines induced distinct plasma antibody responses despite similar B cell receptor repertoires in infant rhesus macaques 

Dear Dr. Fouda:

I'm pleased to inform you that your manuscript has been deemed suitable for publication in PLOS ONE. Congratulations! Your manuscript is now with our production department. 

Kind regards, 

on behalf of

Dr. Mrinmoy Sanyal 

Academic Editor

PLOS ONE